# Nitrogen-Doped Carbon Quantum Dots for Biosensing Applications: The Effect of the Thermal Treatments on Electrochemical and Optical Properties

**DOI:** 10.3390/molecules28010072

**Published:** 2022-12-22

**Authors:** Francesco Ghezzi, Riccardo Donnini, Antonio Sansonetti, Umberto Giovanella, Barbara La Ferla, Barbara Vercelli

**Affiliations:** 1Istituto per la Scienza e Tecnologia dei Plasmi, CNR-ISTP, Via Cozzi 53, 20125 Milano, Italy; 2Istituto di Chimica della Materia Condensata e di Tecnologie per l’Energia, CNR-ICMATE, Via Cozzi 53, 20125 Milano, Italy; 3Istituto di Scienze del Patrimonio Culturale, CNR-ISPC, Via Cozzi 53, 20125 Milano, Italy; 4Istituto di Scienze e Tecnologie Chimiche “Giulio Natta”, CNR-SCITEC, Via Alfonso Corti 12, 20133 Milano, Italy; 5Dipartimento di Biotecnologie e di Bioscienze, Università degli Studi di Milano-Bicocca, Piazza della Scienza 2, 20126 Milano, Italy

**Keywords:** nitrogen-doped carbon quantum dots, hydrothermal synthesis, thermal treatment, optical and electrochemical properties

## Abstract

The knowledge of the ways in which post-synthesis treatments may influence the properties of carbon quantum dots (CDs) is of paramount importance for their employment in biosensors. It enables the definition of the mechanism of sensing, which is essential for the application of the suited design strategy of the device. In the present work, we studied the ways in which post-synthesis thermal treatments influence the optical and electrochemical properties of Nitrogen-doped CDs (N-CDs). Blue-emitting, N-CDs for application in biosensors were synthesized through the hydrothermal route, starting from citric acid and urea as bio-synthesizable and low-cost precursors. The CDs samples were thermally post-treated and then characterized through a combination of spectroscopic, structural, and electrochemical techniques. We observed that the post-synthesis thermal treatments show an oxidative effect on CDs graphitic N-atoms. They cause their partially oxidation with the formation of mixed valence state systems, [CDs]^0+^, which could be further oxidized into the graphitic *N*-oxide forms. We also observed that thermal treatments cause the decomposition of the CDs external ammonium ions into ammonia and protons, which protonate their pyridinic N-atoms. Photoluminescence (PL) emission is quenched.

## 1. Introduction

Carbon quantum dots (CDs) are a new class of carbon-based nanomaterials, which were casually discovered in 2004 during the electrochemical purification of single-walled carbon nanotubes [1]. They could be considered 0-dimensional quasi-spherical nanoparticles with sizes up to 10 nm [2,3] that exhibit photoluminescence (PL) [4]. CDs have gained significant growth within scientific and technology circles because of their peculiar properties that range from excitation wavelength-dependent PL [5] to good biocompatibility [6], high quantum yield (QY), abundance of precursors in nature, excellent aqueous solubility [7] and stability [8], low levels of toxicity [9], chemical stability, resistance to photobleaching [10] and easy modification [11]. In particular, the biocompatibility of CDs is mainly ascribed to the abundant functional groups (hydroxyl, carboxyl, amino, and epoxy) present on their surfaces, which could also act as binding centers for inorganic and organic moieties [12,13,14,15]. Due to the above-mentioned properties, CDs were selected as smart nano-agents for the potential application in various biomedical fields [16,17,18,19,20]. The recent literature works report the rapid development of CDs for practical employment in terms of bioimaging of various cells and animal models [21], diagnostic biosensing (optical and electrochemical biosensors) [22,23], therapeutics (chemotherapy, photodynamic therapy (PDT), photothermal therapy (PTT), multimodal therapy) [24].

Concerning the sensing applications, we have to consider that a biosensor is usually known to consist of three components: a detector, which recognizes the target analyte signal; a converter, which converts the detected signal into a useful output; and a signal processor, which analyzes and processes the output signal [25]. In most sensing systems, CDs function as converters, i.e., they convert intangible information into useful signals recognizable by the signal processor. In an electrochemical biosensor, CDs are generally directly loaded onto an electrode surface, and the detection/analysis of the target analytes is performed through the collection of the altered signals (i.e., current) caused by the interaction of CDs with biomolecules [22]. On the other side, for the design of a CDs-based optical biosensor, the mechanism of sensing has to be taken into account for the application of the suited strategy [6]. From these considerations, it is evident that the knowledge of the ways in which the post-synthesis treatments may influence the optical, structural, and electrochemical properties of CDs could be helpful for the suitable design and development of a CDs-based biosensor. Simple variations in the optical (PL-emission or UV–vis absorption) or electrochemical (redox potential or centers) properties of CDs may cause changes in their mechanism of sensing, which may affect the design strategy of the device.

In a previous work [26] we reported a study on the optical and electrochemical properties of blue-emitting Nitrogen-doped CDs (N-CDs), with particular care in the determination of the redox centers of the samples and the influence of the pH on their optical and electrochemical properties. Starting from those considerations in the present work, we report a study on the ways in which those properties are influenced by post-synthesis thermal treatments. The temperature-dependence of the CDs photoluminescence emission (PL-emission) was studied by Gan et al. [27]; however, to the best of our knowledge, there are no studies on the influence of thermal post-synthesis treatments on the CDs redox centers. Thus, in the present work, blue-emitting N-CDs samples were prepared through a hydrothermal route employing citric acid and urea as bio-synthesizable and low-cost precursors. Two equal amounts of the purified sample were thermally treated for 1 h (N-CDs-160-1) and 8 h (N-CDs-160-8), respectively, and characterized through spectroscopic, structural, and electrochemical techniques. We observed that prolonged post-synthesis thermal treatments affect the electrochemical responses of their graphitic N-atoms, causing the decomposition of the external ammonium groups and the protonation of the pyridinic N-atoms. These results were analyzed and discussed through a combination of spectroscopic, structural, and electrochemical determinations.

## 2. Results

### 2.1. Structural and Optical Characterization

***X-ray diffraction***: The structure of the two samples was characterized by XRD determinations. As previously reported [26], the XRD diffractogram of N-CDs-160-1 suggests that the sample consists of both crystalline and amorphous phases of carbon [28]. In Figure 1a inset, it is possible to note a broad band between 20° and 35°, along with two sharp peaks at 26.8° and 29.0°, respectively (starred in Figure 1a inset). The diffractogram of N-CDs-160-8 shows a broad band around 27° (Figure 1a), which indicates the prevalence of a disordered graphitic structure [29]. The two sharp peaks are still present, but their intensities are much lower than the ones of N-CDs-160-1 (Figure 1a inset). It seems that prolonged thermal treatments may favor the prevalence of amorphous phases of carbon [30] in N-CDs samples.

***FT-IR spectroscopy***: FT-IR spectroscopy was employed to study the nature of the functional groups outside the carbogenic core of the two samples: FT-IR spectra (Figure 1b) show that the functional groups bound at the edges of the basal planes of the graphitic structure of both samples are mainly amides and ammonium carboxylates [26]. In particular, the amide bands are the N-H stretching at 3400 cm^−1^, the C=O stretching at 1680 cm^−1^ and the N-H bending at 1472 cm^−1^; the carboxylate bands are the COO^-^ asymmetric and symmetric stretching at 1570 cm^−1^ and 1400 cm^−1^, respectively, and the ammonium bands are the N-H stretching at 3160 cm^−1^ and the N-H bending at 1389 cm^−1^ [26,31]. The FT-IR spectrum of N-CDs-160-8 shows a further shoulder at ca 2750 cm^−1^ (starred in Figure 1b), which may be assigned to the N-H stretching of protonated pyridine [31]. We suppose that prolonged thermal post-treatment is expected to cause the decomposition of the amide and ammonium terminal groups. In particular, the amide groups may hydrolyze, while ammonium ones may decompose into ammonia (gas) and protons, which are expected to protonate the pyridine N-atoms present in the N-CDs honeycomb matrix (Figure 1c). We excluded the possible protonation of the carboxylate groups, because if we consider the acid dissociation constants *pK_a_* of pyridine (4.2) and an aromatic carboxylic acid, such as benzoic acid (5.2), it results that the protonation of pyridine is ten times more likely to occur. To prove the hypothesis described above, we thermally treated sample N-CDs-160-8 at 80 °C for one week in an aerated oven. Its FT-IR spectrum (Figure 1b) shows that the 2750 cm^−1^ band increases in intensity and broadens. The ammonium band at 3160 cm^−1^ decreases in intensity, while the amide bands at 3400 cm^−1^ and 1680 cm^−1^ remain almost unchanged. Thus, the present results confirm that prolonged thermal post-treatment of N-CDs may cause the decomposition of the ammonium ions and the protonation of pyridinic N-atoms.

***XPS spectroscopy***: XPS spectroscopy was employed to study the chemical composition of the two samples. The survey spectra of both N-CDs samples present three peaks centered at 285.6 eV, 400.6 eV, and 531.6 eV, which were, respectively, assigned to C_1s_, N_1s_, and O_1s_. The high-resolution spectra of C_1s_ and O_1s_ peaks of N-CDs-160-1 and N-CDs-160-8 present the same components, which were deeply discussed for N-CDs-160-1 [26] and do not present marked reciprocal variations after thermal treatment. It is interesting to compare the high-resolution spectra of the N_1s_ peak (Figure 1d) of both samples. As previously reported [26,30], the N_1s_ peak can be deconvoluted into four components, centered at 399.6 eV, 400.6 eV, 401.6 eV, and 402.7 eV, which were assigned, respectively, to pyridinic, pyrrolic, graphitic, and pyridine N-oxide N-atoms (Figure 1c). The estimated relative intensity ratio N_1s_/C_1s_ (Table 1) show that the nitrogen content decreases with the thermal post-treatment.

It is also noteworthy that while the relative percentages of pyridine N-oxide N-atoms remain almost unchanged, the ones of pyridinic, pyrrolic, and graphitic N-atoms vary with the thermal post-treatment. In particular (Table 1), the relative percentage of pyridinic N-atoms decreases from 22% of N-CDs-160-1 to 20% of N-CDs-160-8, probably because of the protonation caused by the thermal decomposition of the ammonium ions, and described above in the FT-IR section. On the other hand, the relative percentages of the graphitic N-atoms also decrease with the thermal post-treatment, passing from 30% in N-CDs-160-1 to 28% in N-CDs-160-8 (Table 1). The literature data [30,32] usually report that higher temperatures of synthesis favor the increase in the graphitic N-atoms. The phenomenon is explained by the increased graphitization induced by the temperature treatment itself. The opposite trend observed in the present case agrees with the above reported XRD determinations which show that in N-CDs-160-8 sample is prevalent a disordered carbon structure. The phenomenon could be explained by the formation of graphitic *N*-oxides [27], and it will be discussed further.

***Optical properties***: The UV–vis absorption spectra of both samples, Figure 2a, show the peaks at 230 nm and 340 nm ascribed, respectively, to the π-π* transition of the aromatic domains, and the n-π* transition (A Figure 2a) of the C=O or C=N [26]. The presence of a third low-energy absorption band (B Figure 2a) indicates that both N-CDs samples are in oxidate state [26,30]. The estimated relative intensity ratios (A_B_/A_A_, see Figure 1) are different (0.03 for N-CDs-160-1 and 0.43 for N-CDs-160-8, respectively), probably due to varying the oxidate state induced by the different thermal treatments. As previously reported [22], after the addition of the reducing agent hydrazine to the water solutions of N-CDs-160-1, the color of the solution turns from blue (oxidized form) to colorless (reduced form) and the A_B_/A_A_ ratio passed from 0.03 to the complete disappearance of the low-energy absorption band. N-CDs-160-8 solutions, after the addition of the same reducing agent, exhibit different behavior (Figure 2b). The solutions’ color passed from blue (oxidized form) to green (intermediate form) and then yellow, Figure 2b inset. The A_B_/A_A_ ratio passed from 0.4 in the oxidate (blue-colored) state to 0.03 in the yellow-colored state (Figure 2b). The A_B_/A_A_ value different from zero, after hydrazine addition, may suggest that sample N-CDs-160-8 is not completely reversibly reduced. It seems that an oxidate component is still present, as will be discussed in the following sections.

Like N-CDs-160-1, the UV–vis responses of N-CDs-160-8 are dependent on the solution pH (Appendix A). By changing the pH from 3 to 9 and vice-versa (through the alternate addition of HClO_4_ and NaOH), the intensity of the A band reversibly increases and decreases, without appreciable variations in its value (Appendix A). On the other hand, the intensity of the B band seems to be not affected by the pH; a reversible ca 30 nm shift to higher/lower energies of its absorption maximum was observed, which is not present in band A.

According to our previous studies on N-CDs-160-1 [26], we ascribed the observed pH-dependence of the UV–vis responses of N-CDs-160-8 to the protonation/deprotonation of its pyridinic-N atoms and we studied its behavior in the in the pH region below 5. As Appendix A shows, it resulted comparable to the one reported for N-CDs-160-1, with similar estimated intrinsic dissociation ($pK_{0}$ = 5.6) and acid dissociation constants ($pK_{a}$ = 2.2). So, like for N-CDs-160-1, in the pH region below 5 the acidity of the protonated pyridinic N-atoms of N-CDs-160-8 increases. Furthermore, the estimated $pK_{0}$ resembles the one of protonated acridine (5.68 [26,33]); thus, the pyridinic N-atoms in the graphitic structure of sample N-CDs-160-8 may be supposed as conjugated acridine systems put at the edges of the honey-comb matrix.

N-CDs-160-1 and N-CDs-160-8 both show PL-emission band peaked at 436 nm (bright blue emission, Figure 2a inset), upon 365 nm excitation with a Stokes shift of ca 100 nm. The full width at half maximum is ca 80 nm, for both samples. The PL quantum yield (PLQY), measured by using quinine sulfate as a reference, is 0.37 for sample N-CDs-160-1, and decreases of ca an order of magnitude (0.07) for sample N-CDs-160-8. It seems that thermal post treatment may cause a decrease in the PLQY of N-CDs, as will be discussed in the following sections.

### 2.2. Electrochemistry

As previously reported [26], sample N-CDs-160-1 presents two irreversible oxidation processes at 0.55 V and 1.44 V (vs. SCE), which were, respectively, assigned to the oxidation of the graphitic and the pyrrolic N-atoms of its honey-comb graphitic structure (Figure 1c). Furthermore, in the acidic pH region (pH ≤ 5), we also observed an irreversible reduction process at ca −1.22 V, which was assigned to the reduction of the protonated pyridine N-atoms.

Sample N-CDs-160-8 in pH 7 buffer solutions shows an open circuit potential E_oc_ = −50 mV. It presents three irreversible oxidation processes at E_ox1_ = 0.35 V, E_ox2_ = 0.66 V and E_ox3_ = 1.46 V (Figure 3, Table 2) and an irreversible reduction one at E_red_ = −0.7 V.

According to the electrochemical oxidation responses of N-CDs-160-1, we assigned the oxidation process at E_ox3_ to the oxidation of the pyrrolic N-atoms. Unlike N-CDs-160-1, we observed that the oxidation and the reduction processes at E_ox1_ and E_red_, respectively, depend on the scan rate and not on its square root, showing that both are not diffusion-controlled. As Table 2 and Figure 4a show, the processes at E_ox1_ and E_red_ are pH dependent.

In particular, E_ox1_ in the pH ranges 3–1 and 5–12 becomes more positive, passing, respectively, from 0.27 V at pH 3 to 0.38 V at pH 1, and from 0.29 V at pH 5 to 0.39 V at pH 12. On the other side, the process at E_red_ becomes more positive in the pH range 3–1, passing from −0.16 V at pH 3 to −0.038 V at pH 1, and it becomes more negative in the pH range 5–12, passing from −0.35 V at pH 5 to −1.22 V at pH 12. In all cases, the pH dependence of the processes at E_ox1_ and E_red_ showed to be linear, see Appendix A and Figure 4a. The electron transfer rate of both processes increases in the acidic pH region (Table 2), and the reduction process becomes reversible at pH ≤ 3 (Figure 4b). On the other side, the oxidation process at E_ox2_ is poorly affected by solution pH and it is characterized by slow electron transfer rates (Figure 4a and Table 2). It is also interesting to note that the potential separation between the oxidation processes at E_ox1_ and E_ox2_ (∆E) is around 0.3–0.4 V and it becomes constant, ca 0.4 V, at pH ≤ 3 (Table 2). A study on the electrochemical responses of stacked tetrathiafulvalene dimers (TTF)_2_ [34] reports that the formation of mixed-valence radical-cations states [(TTF)_2_]^+**·**^ in the dimer stacked system causes the splitting of the TTF first oxidation process into two processes with a potential separation of 0.2 V. In another electrochemical study on the realization and properties of α,ω-capped esathiophene dimer films [35], we observed that the formation of mixed-valence radical-cation states in the film is characterized by a potential separation value (0.4 V) which is comparable with the above-reported ∆E one of the N-CDs-160-8 samples. Starting from these considerations, we may suggest that the processes at E_ox1_ and E_ox2_ may be assigned to the oxidation of the graphitic N-atoms of N-CDs-160-8, which at the open circuit potential of −50 mV may be considered in a partially oxidated mixed valence state, [N-CDs-160-8]^0+^.

Furthermore, like N-CDs-160-1 [26], the electrolysis of N-CDs-160-8 at 0.8 V revealed the progressive decrease in the absorption bands at 340 nm and 630 nm with the parallel emergence of a new band at ca 400 nm (Appendix A). At the same time, the color of the solution turns from blue to dark green and finally to gold yellow. The total charge involved in the oxidation process is 5.2 C. Compared with the total charge required for the oxidation of the graphic N-atoms of N-CDs-160-1, the proposed one looks to be about 30% less. The present results suggest that, like N-CDs-160-1, the graphitic N-atoms of N-CDs-160-8 are fully oxidized to graphitic *N*-oxides, NO-CDs. The electrochemical response of NO-CDs shows an irreversible oxidation process at ca 1.6 V vs SCE (Appendix A, inset 2), which is the already observed oxidation of the pyrrolic N-atoms that were not involved in the electrolysis process.

Thus, if we assume that in the N-CDs-160-8 sample the positive charge of the mixed valence state [N-CDs-160-8]^0+^ could be balanced by the carboxylate ions left free from the ammonium ones (which were thermally decomposed), and we consider an ideal pH = 0 solution (Figure 4a and Appendix A), we may propose the following electrochemical processes for [N-CDs-160-8]^0+^ system:(1)$${\left[N-CDs-160-8\right]}^{0+}{\left(-COO\right)}^{-}+e^{-}\to {\left[N-CDs-160-8\right]}^{00}+-COO^{-}\;E_{0}\;=\;0.02\;V,$$
(2)$${\left[N-CDs-160-8\right]}^{0+}{\left(-COO\right)}^{-}-e^{-}\to {\left[N-CDs-160-8\right]}^{++}+-COO^{-}\;E_{0}\;=\;0.43\;V,$$
(3)$${\left[N-CDs-160-8\right]}^{++}-2e^{-}\to {\left[N-CDs-160-8\right]}^{2+2+}\;\;\;\;\;\;\;\;\;\;\;\;\;\;\;\;\;\;\;\;\;\;\;\;E_{p}\;=\;0.80\;V,$$
(4)$${\left[N-CDs-160-8\right]}^{2+2+}+2H_{2}O\to 2\left(NO-CDs\right)+4H^{+},$$
i.e., ${\left[N-CDs-160-8\right]}^{0+}{\left(-COO\right)}^{-}$ system with one electron could be reduced to the ${\left[N-CQDs-160-8\right]}^{00}$ system and oxidized to the ${\left[N-CDs-160-8\right]}^{++}$ system. The ${\left[N-CDs-160-8\right]}^{++}$ system could be further oxidized with two electrons to the ${\left[N-CDs-160-8\right]}^{2+2+}$ system that after the reaction with two molecules of water and the loss of four protons gives the fully oxidized NO−CQDs form. The observed increased electron transfer rate of ${\left[N-CDs-160-8\right]}^{0+}$ oxidation and reduction processes and the reversibility of the reduction process in the acidic pH region (Table 2) could be explained by the synergic action of the protons coming from both its protonated pyridinic N-atoms (estimated $pK_{a}$ = 2.2) and from the acidic medium that also balance the ${\left(-COO\right)}^{-}$ groups.

## 3. Discussion

The results reported above enable some considerations on the influence of the thermal post-treatment on the optical and electrochemical properties of the N-CDs samples. If we consider the UV–vis spectrum of sample N-CDs-160-8 treated with hydrazine (Figure 5) we can see that it can be deconvoluted into two components: one at 340 nm (red line in Figure 5) and another one at ca 400 nm (blue line in Figure 5). Thus, it seems that two species are present in the N-CDs-160-8 samples.

The first one at 340 nm may be ascribed to the oxidated mixed valence state [N-CDs-160-8]^0+^ that could be reversibly reduced to the [N-CDs-160-8]^00^ state or fully oxidized to the oxidated NO-CDs system. The second component at 400 nm may indicate that a small percentage (ca 15%) of graphitic *N*-oxides is already present in the N-CDs-160-8 structure. NO-CDs show an absorption band at ca 400 nm (Appendix A) and their water solutions are yellow-colored, like the ones of N-CDs-160-8 after hydrazine treatment (Figure 2b inset). The estimated intensity ratio A_B_/A_A_ value different from zero of N-CDs-160-8 solutions treated with hydrazine also show that an oxidized component, which could not be reversibly reduced, is present in its structure. If we consider the electrolysis process for the conversion of the graphitic N-atoms of N-CDs into graphitic *N*-oxides, we see that the total charge requested for the full electrolysis of the graphitic N-atoms of N-CDs-160-8 is ca 30% lower than the one requested for the same oxidation process of N-CDs-160-1, as if part of them could be already oxidized. The XPS determinations reported above show that the percentage of the graphitic N-atoms of N-CDs-160-8 is lower than the one of N-CDs-160-1 (Table 1) consistent with the proposed hypothesis. Thus, we may suggest that a prolonged thermal post-treatment may have an oxidative action on the graphitic N-atoms of N-CD samples. Concerning the present case, it causes the first conversion of the graphic N-atoms into the oxidate mixed valence state [N-CDs-160-8]^0+^, which in small percentage could be further oxidized into graphitic *N*-oxides.

Furthermore, the observed PL-quenching with temperature may be ascribed to their oxidation. The literature reports that PL-quenching of CDs occurs when they are oxidized, probably because of the trapping of the excited electrons by non-radiative defects [36].

## 4. Materials and Methods

All used chemicals and solvents were reagent-grade and employed as received. pH buffers (1, 2, 3, 4, 5, 7, 9, 12) for UV–vis and electrochemical analyses were purchased by Carlo Erba.

### 4.1. Nitrogen-Doped Carbon Quantum Dots Synthesis

Nitrogen-doped CDs (N-CDs) were prepared according to a previous reported route [37]. An amount of 2 g (10 mmol) of citric acid and 1.9 g (30 mmol) of urea were dissolved in 50 mL of deionized water, transferred into a Teflon-lined stainless steel reactor (100 mL) and heated at 160 °C for 4 h. After the reactor was cooled to room temperature naturally, the solution was filtrated through a microporous membrane (0.22 μm) to remove the large particles and precipitated with EtOH followed by centrifugation. The re-dispersion/precipitation procedure was repeated three times to ensure the effective removing of precursor residues and small derivatives. Two equal amounts of the purified sample were thermally treated in an aerated oven at 60 °C for 1 h (N-CDs-160-1) and 8 h (N-CDs-160-8), respectively, to study the influence of the thermal post-treatment period on N-CDs properties. Both dried samples are blue-colored solids soluble in water. The solids were stored in closed vials at 4 °C, and showed to be stable for several months. The synthesis route was repeated three times, and the good reproducibility of the properties of the obtained materials proved its reliability. Furthermore, the ninhydrin test was performed on N-CDs samples to exclude the presence of primary amine-containing residues. The reaction yield was ca 45 % in all cases.

### 4.2. Characterization Techniques

UV–vis spectra were collected with a Perkin Elmer Lambda 35 spectrometer; FT-IR spectra of KBr pellets were recorded on a Nicolet 6700 spectrometer, 64 scans, resolution 4 cm^−1^.

The continuous wave PL spectra were recorded by liquid nitrogen-cooled charge-coupled device combined with a monochromator, and excited using a monochromated xenon lamp. PL-QYs in solutions were measured by comparison with quinine in 1N H_2_SO_4_ water solution as a standard.

X-ray diffraction (XRD) spectra were recorded using a Siemens D500 X-ray diffractometer with Cu-Kα radiation (λ = 0.15418 nm) and collected in the 2Θ range of 10°–40°, with a step size of 0.01° and 5 sec/step.

The XPS apparatus consisted in a VSW CLASS 100 hemispherical analyzer, equipped with a single channel detector, with a non-monochromated Al/Mg X-ray source (VSW model TA10). The apparatus is equipped with an impact ionization ion gun VG EX05 with a raster scan unit (SAX 346) for imaging and depth profiling. Ar, Ne and He gas can be used. The core level fits have been performed through a Voigt profile including a Lorentzian function (accounting for lifetime broadening) and a Gaussian function (accounting for the finite instrumental resolution). The fitting routine also included a Shirley background, mimicking the secondary electron background. C_1s_ line shape was instead fitted with a Doniach–Sunjic’s curve using an asymmetry factor, typically of 0.05. Atomic sensitivity factors were specifically determined for the apparatus using the Lindau’s cross sections [38].

Electrochemistry was performed at room temperature in three electrode cells; counter electrode was platinum; reference electrode was SCE. Working electrodes were glassy carbon (GC 0.06 cm^2^). Supporting electrolyte was 0.1 M NaClO_4_. Cyclic voltammetry (CV) was performed at a scan rate of 0.1 V s^−1^. The voltammetric apparatus was Metrohm Autolab 128 N potentiostat/galvanostat. CVs were performed in 1 mg mL^−1^ N-CDs solutions.

## 5. Conclusions

In this work, we studied the effects of the post-synthesis thermal treatment on the optical and electrochemical properties of N-CDs. From the combined analysis of the UV–vis and CVs responses, we propose that the N-CDs samples thermally treated for 8 h may be composed of two oxidized species. The most abundant component (ca 86%) was ascribed to the oxidated mixed valence state [N-CDs-160-8]^0+^ that could be reversibly reduced. The less abundant one (ca 15%) was attributed to the presence of graphitic *N*-oxides in the N-CDs structure, which are not reversibly reduced. The presence of NO-CDs in N-CDs-160-8 samples is supported both by UV–vis and XPS determinations. The former show that the water solutions of N-CDs-160-8 treated with hydrazine are yellow-colored, like NO-CDs ones, and their estimated intensity ratio A_B_/A_A_ value is different from zero, which shows that in N-CDs-160-8 oxidized species are present that could not be reversibly reduced. On the other hand, XPS data present a decrease in the relative percentage of the graphitic N-atoms passing form sample N-CDs-160-1 to sample N-CDs-160-8, probably because of the NO-CDs species already present in the structure of the latter. Thus, we may suggest that a prolonged post-synthesis thermal treatment of N-CDs samples may have an oxidative effect on N-CDs. It causes the first conversion of their graphic N-atoms into the oxidate mixed valence state [N-CDs-160-8]^0+^, which in small percentage could be further oxidized into graphitic *N*-oxides. Furthermore, FT-IR data show that a prolonged thermal treatment seems to also cause the decomposition of the ammonium ions, present the edges of the basal planes of the graphitic structure of N-CDs samples into ammonia and protons that protonate their pyridinic N-atoms. This result is supported by the XPS data that show a decrease in the relative percentage of the pyridinic N-atoms in the N-CDs sample thermally treated for 8 h. Finally, the temperature may cause the activation of more non-radiative channels; thus, the excited electrons in CDs reversed back to the ground state by non-radiative processes, leading to PL-quenching.

We believe that the results reported above may help in realizing CDs-based biosensors. We expect that with a simple thermal treatment of N-CDs, it could be possible to vary their mechanism of sensing and enable multiple designs of devices that employ the same CDs sample but are realized according to different strategies. Further studies are still in course to verify this hypothesis.

## Figures and Tables

**Figure 1 molecules-28-00072-f001:**
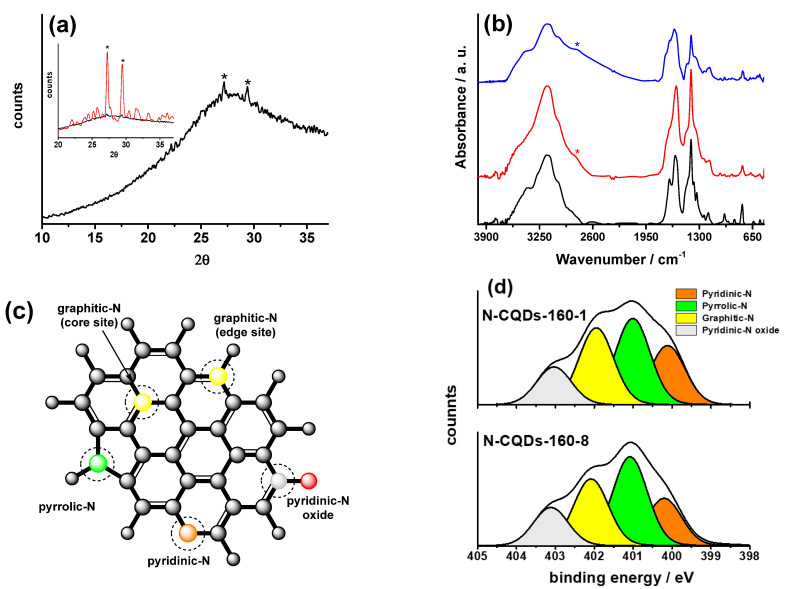
(**a**) XRD pattern of N-CDs-160-8. Inset XRD of sample N-CDs-160-1 (red line) and N-CQDs-160-8 (black line); (**b**) FT-IR spectra of samples N-CDs-160-1 (black-line), N-CDs-160-8 (red line) and N-CDs-160-8 thermally treated at 80 °C for 1 week (blue line); (**c**) structure of common types of N-doping/functionalizations in N-CDs; (**d**) XPS high-resolution spectra of N_1s_ of N-CD samples.

**Figure 2 molecules-28-00072-f002:**
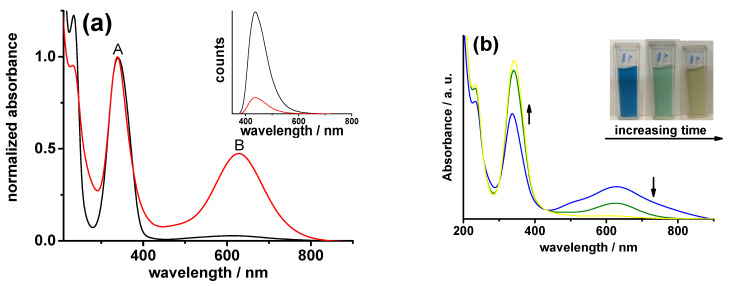
(**a**) Normalized UV–vis absorption spectra of 1 mg mL^−1^ water solutions of N-CDs-160-1 (black line) and N-CDs-160-8 (red line); inset: PL spectra at 365 nm excitation of N-CDs-160-1 (black line) and N-CDs-160-8 (red line). (**b**) UV–vis absorption spectra of sample N-CQDs-160-8 upon hydrazine addition at increasing time; inset: photoghaph of sample N-CQDs-160-8 solutions after hydrazine addition at increasing time of reaction (blue oxidized form at t = 0 h, yellow reduced form at t = 70 h, green an intermediate form).

**Figure 3 molecules-28-00072-f003:**
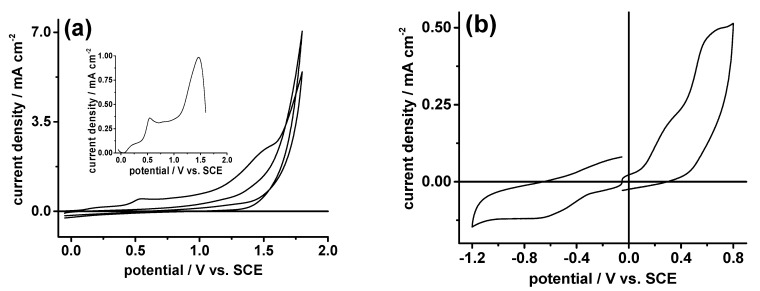
(**a**) Cyclic voltammogram of 0.5 mg ml^−1^ N-CDs-160-8 and CG electrode in pH 7 buffer + 0.1 M NaClO_4_. Scan rate: 0.1 V s^−1^. Inset: Single-sweep voltammogram of N-CDs-160-8 CG background subtracted. (**b**) Cyclic voltammograms of 1 mg ml^−1^ of N-CDs in pH 7 buffer + 0.1 M NaClO_4_. Scan rate: 0.1 V s^−1^.

**Figure 4 molecules-28-00072-f004:**
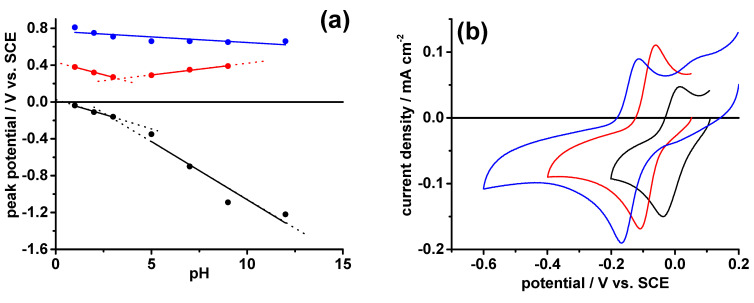
(**a**) The linear relationships of oxidation (E_ox1_ red line, E_ox2_ blue line) and reduction (black line) peak potentials of N-CDs-160-8 versus pH. (**b**) Cyclic voltammograms of N-CDs-160-8 in pH 1 (black line), pH 2 (red line) and pH 3 (blue line) buffers + 0.1 M NaClO_4_. Scan rate: 0.1 V s^−1^.

**Figure 5 molecules-28-00072-f005:**
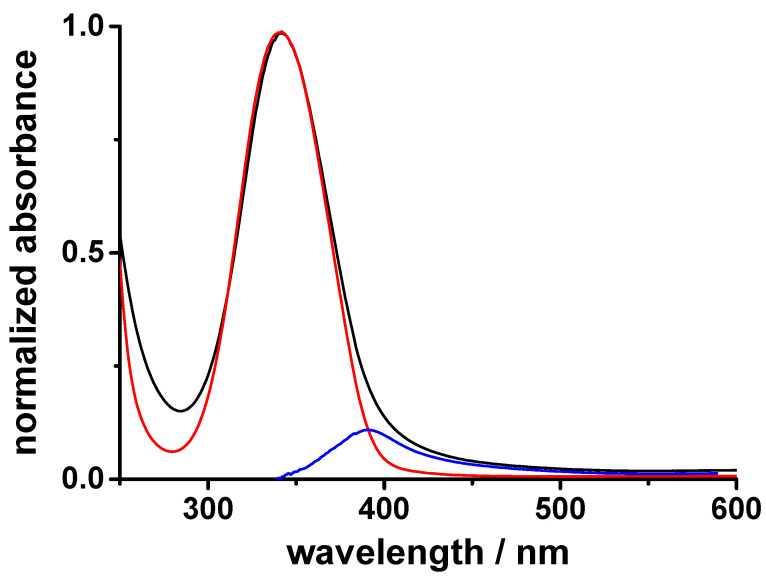
UV–vis absorption band A of 1 mg mL^−1^ water solution of N-CDs-160-8 fully reduced with hydrazine (black line); [N-CDs-160-8]^0+^ component (red line); NO-CDs component (blue line).

**Table 1 molecules-28-00072-t001:** XPS peak intensity ratio N_1s_/C_1s_ and relative percentages of graphitic, pyrrolic, pyridinic and pyridine N-oxide (% N_gr_, % N_pyr_, % N_py_, % N_py-O_, respectively).

Sample	N_1s_/C_1s_	% N_gr_	% N_pyr_	% N_py_	% N_py-O_
**N-CQDs-160-1**	0.18	30	33	22	15
**N-CQDs-160-8**	0.16	28	37	20	15

**Table 2 molecules-28-00072-t002:** Oxidation and reduction peak potentials vs. SCE (E_oxn_ and E_red_/V), peak width (∆E_oxn_ (∆E_red_/mV) and E_0x1_ E_ox2_ potential separation (∆E/V) for sample N-CQDs-160-8 at different pH values.

pH	E_ox1_; E_ox2_	E_red_	∆E_pox1_; ∆E_pox2_	∆E_pred_	∆E
**1**	0.38 *; 0.81 *	−0.038 *	40; 103	53	0.43
**2**	0.32 *; 0.75 *	−0.11 *	50; 100	41	0.43
**3**	0.27 *; 0.71 *	−0.16 *	50; 120	59	0.44
**5**	0.29 *; 0.66 *	−0.35 *	90; 140	64	0.37
**7**	0.35 *; 0.66 *	−0.70 *	100; 150	172	0.30
**9**	0.39 *; 0.65 *	−1.09 *	150	320	0.30
**12**	0.66 *	−1.22 *	230	240	

* Estimated values.

## Data Availability

Data is contained within the article and Appendix A.

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
