# Peer review of "Nitrogen-Doped Carbon Quantum Dots for Biosensing Applications: The Effect of the Thermal Treatments on Electrochemical and Optical Properties"

_molecules, 2022, doi:10.3390/molecules28010072_

Round 1

Reviewer 1 Report

The manuscript is well developed and is a continuation of the research of the authors [22], which significantly supplements the cited work. Changes in the characteristics of the material during heat treatment are demonstrated, which is of fundamental importance for the application of CD. Considering the interest that the authors' work [22] aroused, there is confidence that the presented manuscript after publication will be in demand by readers and well cited. The manuscript describes in detail all the necessary information about the experimental technique, equipment. The experimental data are reasonably interpreted.

Notes are minor

1. It is necessary to provide a literary reference to the data in accordance with which the bands in the IR spectra were assigned.

2. How stable is the obtained material during storage and how do its characteristics (optical, electrochemical) change in this case.

Author Response

see the atteched file

Reviewer 2 Report

In this article the author reports ‘Nitrogen-doped Carbon Quantum Dots for Biosensing applications: The Effect of The Thermal Treatments on Electrochemical and Optical Properties’. This topic will be surely interesting for many researchers working in the related fields. However, there are some points that need to be addressed before possible publication.

 Recommendation: Major as noted

     1.      The novelty of the present article should be discussed a little more in the Introduction section.

2.      A separate subheading should be added for the synthesis of CDs in the 4. Materials and Methods section.

3.      Did the authors perform dialysis of the QDs? Dialysis is very important for the purification of QDs. It should be mentioned in the synthesis section.

4.      The discussion in the FTIR analysis section should be properly cited with relevant references.

5.      Page 4, line 145-147 should be cited with relevant references.

6.      The author should write the purpose for each test in one/two sentences (in brief) before explaining the results of the characterization techniques. Therefore, the logic and organization of this part will be enhanced.

7.  The formatting and grammatical errors in the article need to be checked carefully.

8.     All equations should be numbered.

9. The authors have cited relevant references in the Introduction section; however the manuscript needs to be highlighted with some recent reports to further broaden the impact, related literatures:      https://doi.org/10.1155/2022/2426749;https://doi.org/10.1016/j.cej.2021.132818; https://doi.org/10.1021/acsapm.2c01579;https://doi.org/10.1016/j.bioelechem.2021.107999;https://doi.org/10.1016/j.electacta.2021.139803;https://doi.org/10.1021/acsabm.2c00664;https://doi.org/10.1021/acsami.1c08111

Round 2

Reviewer 2 Report

The authors have addressed all the questions raised before therefore the manuscript can be accepted in the present form